# Knowledge, attitudes and practices relating to HIV self-testing following its introduction in the Bas-Sassandra region of Côte d'Ivoire: the case of the ATLAS project

Arlette Simo Fotso[1,2]*, Christian Koukobo[2], Romain Silhol[3], Arsène Kouassi Kra[2], Marie-Claude Boily[3], Anthony Vautier[4], Joseph Larmarange[1,2]* on behalf of the ATLAS team¶

**1** Institut National des Études Démographiques (Ined), France, **2** Centre Population et Développement (Ceped), Université Paris Cité, Université Sorbonne Paris Nord, Institut de Recherche pour le Développement (IRD), Inserm, France, **3** MRC Centre for Global Infectious Disease Analysis, School of Public Health, Imperial College, London, United Kingdom, **4** Solidarité Thérapeutique et Initiatives pour la Santé (Solthis), France

¶ Membership of the ATLAS team is provided in the Acknowledgments.
* joseph.larmarange@ird.fr (JL); arlette.simo-fotso@ined.fr (ASF)

## Abstract

### Background

Awareness of HIV status is crucial for accessing HIV care and prevention but remains suboptimal in West Africa. The ATLAS initiative, launched in Côte d'Ivoire, Mali, and Senegal, addressed this gap by distributing approximately 380,000 HIV self-testing (HIVST) kits from 2019 to 2021, primarily to key populations and their social networks. This study assessed levels and correlates of Knowledge, Attitudes, and Practices (KAP) related to HIVST in the Bas-Sassandra region of Côte d'Ivoire following ATLAS's introduction.

### Method

A cross-sectional population-based survey was conducted in the Bas-Sassandra region in 2021 among individuals aged 15–49. A total of 6,271 people (3,203 men and 3,068 women) were interviewed. They were selected using a three-stage stratified sampling approach in the Bas-Sassandra region. Bivariate statistics and multivariable logistic regressions were used to assess KAP levels and the associated factors.

### Results

Although few participants reported having heard about HIVST (11%) or having used it (3%), most of them reported that if it were freely available, they would be interested/very interested in using it for themselves (76%), as well as for their sexual partners

**Data availability statement:** A minimal anonymized dataset and an R script allowing to replicate the presented analyses have been made publicly available on Zenodo (https://doi.org/10.5281/zenodo.15064708).

**Funding:** "This analysis was supported by Unitaid (grant number 2018–23 ATLAS) through a collaborative agreement with Solthis. The funders had no role in study design, data collection and analysis, decision to publish, or preparation of the manuscript.".

**Competing interests:** The authors have declared that no competing interests exist.

(75%). Education and wealth were positively associated with knowledge and positive attitudes towards HIVST among both men and women, whereas age was positively correlated to knowledge and use of HIVST among men only. The number of sexual partners over the last 12 months was positively associated with knowledge of HIVST and willingness to use HIVST for themselves or their sexual partners among both sexes. We also found that high HIV-related knowledge and low levels of negative attitude were positively associated with positive attitudes towards HIVST, while exposure to the media appeared to be correlated to knowledge of HIVST.

## Conclusion

The high level of positive attitudes towards HIVST calls for a scaling up of access to HIVST in the region. Specific attention to groups with the worst KAP, such as the less educated, the poor or those with more HIV-related negative attitude, could enhance the success of such initiatives.

---

## 1. Introduction

HIV testing is a central element in the strategy to end the HIV epidemic, as awareness of HIV status is the entry point to HIV care and prevention. In Western and Central Africa, in 2023, only 81% of people living with HIV (PLHIV) were aware of their HIV status [1]. With an HIV prevalence of 2.2% among adults in 2023, Côte d'Ivoire is one of the Western African countries most affected by the HIV epidemic. However, in this country only 82% of PLHIV knew their status in 2023 [2]. This is well below the UNAIDS 2025 target of reaching 95% of people PLHIV who know their HIV status [1], thus highlighting the presence of obstacles linked to the classic testing strategies and the need to improve access to HIV screening services through innovative technologies such as HIV self-testing (HIVST).

Recommended by the WHO since 2016 as a complement to classic screening strategies, HIVST is a new medical tool that reduces barriers to access to screening [3], especially in contexts characterized by high levels of HIV-related discrimination and stigma [1,4].

In order to help fill the testing gap through HIVST, and build on the STAR project in Southern and Eastern Africa [5], the ATLAS programme (*AutoTest de dépistage du VIH: Libre d'Accéder à la connaissance de son Statut*) received funding from the global health initiative UNITAID, through the NGO *Solidarité Thérapeutique et Initiatives pour la Santé* (SOLTHIS) and the French National Research Institute for Sustainable Development (IRD), in partnership with national authorities and civil society organizations. The aim of the programme was to introduce, implement and scale up HIVST in West Africa. The ATLAS initiative has distributed around 380,000 HIVST kits through several distribution channels between 2019 and 2021 in Côte d'Ivoire, Mali and Senegal [6]. More than half of these kits (~200,000 kits) were distributed in Côte d'Ivoire. Alongside the ATLAS programme, UNITAID has also funded MTV Shuga Babi, a two-season multimedia campaign, broadcast in Côte d'Ivoire in 2019–2021,

targeting mainly young people aged 15–24, promoting messages consistent with national AIDS strategies and highlighting existing resources. Additional details about the two programmes are provided in the methods section.

Several studies have shown that, for many users, HIVST promotes discretion and autonomy, reaches first-time testers, and is associated with an increase in testing frequency [7–15]. HIVST is very well accepted, particularly by key populations and vulnerable groups, and reaches people living with HIV [16–20]. In addition, recent studies have demonstrated the positive impact of HIVST distribution on HIV diagnosis, HIV treatment coverage, HIV incidence, and HIV-related mortality at population level [21,22]. But evidence on knowledge of, attitudes towards and practices relating to HIVST remains limited in the region [19,23].

The Shuga Babi and ATLAS programmes conducted a joint, original, population-based, cross-sectional survey in the Bas-Sassandra District of Côte d'Ivoire to add new data to the limited existing empirical evidence on knowledge of, attitudes towards and practices relating to HIVST (KAP) in Africa. In this study, we aimed to use this survey to assess the levels and correlates of knowledge of, attitudes towards, and practices relating to HIV self-testing following the introduction of HIVST by the ATLAS project in Côte d'Ivoire. Identifying correlates of KAP could provide additional evidence-based tools for policymakers, which would enable them to implement actions to improve HIVST uptake and, ultimately, HIV status awareness.

## 2. Method

### 2.1. Data

We used data from a household survey that was carried out in the Bas-Sassandra district in Côte d'Ivoire: the Shuga Babi - ATLAS Household Survey (SBAHS).

**2.1.1. Study setting.** The Bas-Sassandra district is located in the southwest of Côte d'Ivoire and is one of the country's fourteen administrative districts. According to the most recent census in 2014, the district had a total population of 1,057,241 people aged 16–49, with 70% living in rural areas. It is one of the wealthiest districts in the country, with the San Pedro port activities and its cocoa and coffee production. These agricultural activities regularly attract a significant number of migrants and seasonal workers, who come to work on farms or in the sex trade [24]. The district is subdivided into 8 sub-districts or departments, 31 sub-prefectures (urban settings) and 533 localities (rural settings).

**2.1.2. Description of the ATLAS programme.** The ATLAS programme distributed 74,785 HIVST kits in the Bas-Sassandra district over the years 2019–2021. More precisely, 42,997, 23,287, and 8,501 kits were distributed in the Soubré, San Pédro, and Tabou health sub-districts of the district respectively. The ATLAS programme mainly relies on two modes of distribution of HIVST kits. The first is the primary distribution strategy, where peer educators issued kits to the primary contacts for their personal use. The second is a secondary distribution, where primary contacts are issued with additional testing kits to redistribute to their sexual and social network – peers, partners and clients. The ATLAS activities are organized through eight different delivery channels: five facility-based approaches for the delivery of HIVST kits through public or community-based healthcare facilities, and three community-based approaches involving outreach activities targeting female sex workers (FSW), men who have sex with men (MSM), and people who use drugs (PWUD) [6]. Approximately 90% of HIV self-testing (HIVST) kits were distributed to key populations. During the implementation period, HIVST kits were not available in pharmacies, and there was no other free distribution programme comparable to ATLAS in the district. The only exception was the United States President's Emergency Plan for AIDS Relief (PEPFAR), which distributed a relatively small number of HIVST kits — around 15,000 — over the same period in the district.

**2.1.3. The MTV Shuga Babi Campaign.** Alongside the ATLAS programme, the MTV Shuga Babi, a two-season multimedia campaign, was broadcast in Côte d'Ivoire over the years 2019–2021, targeting mainly young people aged 15–24. The first season of MTV Shuga Babi campaign combined issues of HIV transmission and prevention messages addressed to young people through entertaining stories tailored for them. The central component of MTV Shuga Babi

 

broadcasts in 2019 was a television (TV) drama series linked to complementary media initiatives to ensure that people with little access to television were reached. Using the same approach, MTV implemented a second season, which was broadcast in 2021. The second season had one episode focusing on the existence and the benefits of HIVST, showing the administration of an HIVST kit on-screen, highlighting the importance of follow-up testing and treatment after a reactive HIVST test and portraying preventive behaviour and repeat testing for those testing negative. The aims of the campaign were simultaneously to raise awareness and to create a demand for HIVST and other prevention strategies among young people.

**2.1.4. Data collection.** The SBAHS survey was conducted between November 20th and December 31st, 2021, about 24 months after the start of the ATLAS project activities (distribution of HIV self-tests), and 6 months after the television broadcast of the second season of the MTV series Shuga Babi. This was a joint survey of the ATLAS and MTV Shuga Babi programmes, aiming, on the basis of the interviewees' reports, to assess the primary and secondary outcomes in the evaluations of the two initiatives. For the ATLAS programme, the survey aimed to quantify baseline HIV testing in the 12 months prior to the survey, HIVST awareness and uptake and links to care as primary and secondary outcomes.

SBAHS was a cross-sectional population-based survey designed to be representative of individuals aged 15–49 living in the study area. It used a three-stage stratified sampling approach. The district was divided into four areas, three corresponding to the three health sub-districts or departments where the ATLAS activities were concentrated (Soubré, San Pédro, Tabou) and the last one corresponding to the other health department. Each area was further divided according to urban/ rural setting, resulting in 8 strata. In order to capture sufficient data in the three ATLAS departments for statistical analyses, the first three strata were over-represented, accounting for two-thirds of the total sample. Similarly, the urban setting was over-represented (about 50% of the sample compared to 30% in the general population) to enable the sample sizes in each setting (urban/rural) to be equal.

Firstly, enumeration areas (EA) were selected for each stratum with a probability proportional to size, resulting in a total of 60 EAs. Second, 60 households per EA were selected using simple random sampling. Third, all *de facto* household members eligible (aged 15–49 years) were invited to participate. Those who gave their written informed consent – or had written informed assent from parents/guardians for minors — were interviewed.

The sample size was calculated to ensure that the study's power was estimated at 92% and 99% to detect a minimum difference of 10% and 15%, respectively, in the ATLAS primary outcome compared to the 2018 PHIA (Population-Based HIV Impact Assessment) survey. The inclusion criteria were individuals aged 15–49 years, de facto household members (i.e., those present in the household at the time of the survey), individuals aged 18 or older with written informed consent, and those aged 15–17 with written informed assent and parent/guardian's written informed consent. The exclusion criteria included individuals younger than 15 or older than 49 years, those not present in the household at the time of the survey, refusal by the participant and/or parent/guardian, refusal by individuals aged 18 or older, refusal by a parent/guardian or lack of assent from individuals aged 15–17, and cognitive issues preventing the individual from providing informed consent.

Face-to-face interviews with a trained interviewer were conducted in French or in one of the main local languages, which are English, Godie, Fanti, Malinke, Kroumen, Bakoue, Baoule, Bete and Neo, using a structured questionnaire loaded on tablets. Ethics approvals were obtained from the Côte d'Ivoire Ministry of Health (N/ref: 051–21/MSHP/ CNESVS-km), the London School of Hygiene and Tropical Medicine (LSHTM) (LSHTM Ethics Ref: 26258), and the World Health Organization (WHO) (ERC.0003596) ethics committees in June, July and September 2021 respectively. The survey was implemented in collaboration with the *École Nationale Supérieure de Statistique et d'Économie Appliquée in Côte d'Ivoire* (ENSEA).

**2.1.5. Inclusivity in global research.** Additional information regarding the ethical, cultural, and scientific considerations specific to inclusivity in global research is included in the Supporting Information (S1 Checklist).

## 2.2. Outcome variables

We defined four dichotomous dependent variables related to knowledge, attitudes, and practices relating to HIV self-testing derived from the questions presented in the supporting information S1 Table. The first dependent variable measured individuals' knowledge of HIVST, assessing whether the respondent had already heard of HIV self-testing, coded as "yes" or "no". The second and third dependent variables concerned the respondents' interest in using HIVST for themselves or for their partners if it was freely available. Each of these were binary variables indicating whether the individuals responded that they interested/very interested or not. These variables reflected the respondents' attitudes towards HIVST. The fourth variable reflecting practices asked individuals whether they had ever used HIVST. This was also a binary variable, coded as "yes" or "no".

## 2.3. Independent variables

We grouped the independent variables used in this analysis into five sets.

**Socio-demographic and economic variables** included sex (men, women), the age group (15–24, 25–34, 35–49 years), the level of education (no formal education, primary, secondary or tertiary education), whether or not currently in a relationship, and the wealth index tercile (poor, neither poor nor rich, rich) derived from the wealth index. The index was calculated using multiple correspondence analysis (MCA) using household asset variables such as household ownership of certain consumer goods (TV, radio, etc.) and housing characteristics (availability of electricity, building materials, type of water supply, type of toilet, and fuel used for cooking).

**Behavioural variables** included the number of sexual partners in the last 12 months (0, 1, 2 or more) and an indicator relating to who made healthcare decisions for the respondent.

**The HIV-related knowledge and attitude** set included two variables. The categorical knowledge variable was based on a series of 16 questions about HIV knowledge (S2 Table). The answers to this question were categorized into three levels: poor, moderate, and good, corresponding to fewer than 10, 10–12, and 13 or more correct answers respectively. The level of negative attitude toward PLHIV was classified as high if the respondent gave a stigmatizing response to at least two of the five questions on HIV-related attitude (see S3 Table), and low otherwise.

**Exposure to the media** was measured by a binary media exposure variable (low, high) based on the frequency of exposure to radio, television, and Internet access. Exposure was high if the individual was exposed to at least two media per week and low otherwise.

**Contextual variables** were also used as potential correlates, such as the place of residence (urban, rural) and the department (Soubré, San Pédro, Tabou and others).

## 2.4. Methodology

We first performed univariate and bivariate analyses to assess levels and factors associated with Knowledge, Attitudes, and Practices (KAP). Univariate analyses involved describing the characteristics of the sample. A cross-tabulation using the Chi² test with Rao & Scott's second-order correction was applied to the bivariate analysis to examine the association between the outcome and independent variables, identifying significant associations at the 5% significance level.

To check whether the associations observed in bivariate analysis remained after controlling for other covariates, a quasi-binomial multivariable logistic regression was performed. Regressions were conducted separately for men and women to assess the KAP correlates in each group. The results are presented as adjusted Odds Ratios (aORs) with 95%-confidence intervals (CI). The latter accounts for model-robust standard errors, adjusting for data clustering. The adjusted Generalized Variance Inflation Factor (GVIF) [25] was used to ensure that there were no serious multicollinearity issues in the multivariate regression. It is important to note that some variables were excluded from all regressions for the multivariate analysis because they were a source of multicollinearity. Analyses were performed with R software (version

4.4.3) using the packages survey, gtsummary and ggstats. Unless otherwise specified, all analyses reported in this study were weighted and took the survey design into account.

A minimal anonymized dataset and an R script allowing to replicate the presented analyses have been made publicly available on Zenodo (https://doi.org/10.5281/zenodo.15064708).

## 3. Results

### 3.1. Participant characteristics

A total of 6,271 people (3,203 men and 3,068 women) were interviewed and completed the questionnaire (Table 1). A large proportion were in a relationship (77%), had no formal education (46%), and were aged 15–24 (36%). Most participants were living in rural areas (62%) and in the department of San Pedro (35%). Most reported that decisions about their own health were taken mainly by a third party (53%), and that they had had only one sexual partner over the 12 months preceding the survey (69%). Half of the population had an average knowledge of HIV, and 63% presented a low level of negative attitude towards PLHIV. Only 39% of the population had high exposure to the media.

With regard to KAP relating to HIVST, although very few participants reported having heard about HIVST (10% of the women and 11% of the men), most of them showed a positive attitude towards HIVST and reported that they would be interested/very interested in using it for themselves (73% of the women and 78% of the men), as well as for their sexual partners (72% of the women and 78% of the men), if it was freely available. About 3% of both women and men reported that they had already used a HIVST in the past.

### 3.2. Factors associated with HIVST knowledge

A bivariate analysis of the association between KAP and the independent variables is presented in Table 2. The first columns show that knowledge of HIVST was significantly and positively associated with age group (p = 0.035), education (p < 0.001), wealth (p < 0.001) and being in a relationship (p = 0.009). Knowledge of HIVST was also significantly higher among people whose decision regarding their own health was taken by themselves and by a third party (p = 0.008), for those with two or more sexual partners (p < 0.001), high HIV-related knowledge (p < 0.001), low HIV-related negative attitude (p < 0.001), and high exposure to the media (p < 0.001).

The multivariable analysis presented in Fig 1 and S4 Table for men and in Fig 1 and S5 Table for women helps to apprehend the persistence of these associations after controlling for other variables. Overall, the tables show that after testing the model and removing the variables that were a source of multicollinearity – the variables *currently in a relationship*, *who makes the decisions about your own health care* and *place of residence* — in the final model there was no strong multicollinearity issue in the regressions, (aGVIF is $\sqrt{5} \approx 2.23$) [26,27].

Among the men, age, education and wealth proved to be positively associated with knowledge of HIVST. Men aged 35–49 years (aOR: 1.78 [95%CI: 1.26 to 2.53]) and 25–34 years (aOR: 1.94 [1.33 to 2.81]), those with secondary or higher education (aOR =2.53 [95% CI: 1.72 to 3.71], those from the highest wealth tercile (aOR:1.92 [95%CI: 1.19 to 3.10]), those with two sexual partners or more (aOR:1.95 [95%CI: 1.12 to 3.05]), those with low levels of negative attitude towards PLHIV (aOR:1.61 [95%CI: 1.15 to 2.26]), and those with high exposure to the media (aOR:1.54 [95%CI: 1.08 to 2.19]) had higher odds of having heard of HIVST compared to reference groups of those aged 15–24 years, with no education, belonging to the poorest wealth tercile, with no sexual partner, those expressing high level of negative attitude toward PLHIV and low media exposure respectively. People from the Tabou had lower odds of HIVST knowledge (aOR:0.45 [95%CI: 0.25 to 0.81]) than those living in other departments. Similar patterns of association were found among women except for age and the negative attitude variables, which appeared non-significantly correlated to HIVST knowledge.

**Table 1. Characteristics of the study population by sex.**

| Characteristics | Women (n = 3068)[1] | Men (n = 3203)[1] | Overall (n = 6,274)[1] |
|---|---|---|---|
| **SOCIO-DEMOGRAPHIC AND ECONOMIC FACTORS** | | | |
| **Age group** | | | |
| 15–24 years old | 38.4% (1,259) | 34.6% (1,192) | 36.4% (2,451) |
| 25–34 years old | 33.0% (953) | 29.7% (913) | 31.4% (1,866) |
| 35–49 years old | 28.6% (856) | 35.7% (1,098) | 32.2% (1,954) |
| **Level of education** | | | |
| none | 55.8% (1,629) | 37.4% (1,205) | 46.4% (2,834) |
| primary | 21.5% (637) | 24.3% (714) | 22.9% (1,351) |
| secondary or higher | 22.7% (802) | 38.3% (1,284) | 30.6% (2,086) |
| **Wealth index** | | | |
| poor | 35.6% (1,029) | 36.2% (1,114) | 35.9% (2,143) |
| neither poor nor rich | 30.0% (884) | 31.8% (959) | 30.9% (1,843) |
| rich | 34.4% (1,155) | 32.0% (1,130) | 33.2% (2,285) |
| **Currently in a relationship** | | | |
| no | 17.0% (575) | 29.3% (1,025) | 23.3% (1,600) |
| yes | 83.0% (2,493) | 70.7% (2,178) | 76.7% (4,671) |
| **BEHAVIOURAL FACTORS** | | | |
| **Number of sexual partners over the last 12 months** | | | |
| 0 partner | 14.7% (447) | 18.2% (632) | 16.5% (1,079) |
| 1 partner | 81.1% (2,468) | 57.4% (1,804) | 69.0% (4,272) |
| 2 partners or more | 4.2% (145) | 24.3% (758) | 14.4% (903) |
| **The person who generally makes the decision about their own health care** | | | |
| themselves | 11.4% (384) | 66.9% (2,080) | 39.7% (2,464) |
| a third party | 79.6% (2,430) | 28.4% (973) | 53.5% (3,403) |
| themselves and a third party | 8.9% (254) | 4.7% (150) | 6.8% (404) |
| **HIV-RELATED KNOWLEDGE AND ATTITUDE** | | | |
| **HIV knowledge** | | | |
| low | 26.3% (816) | 18.6% (603) | 22.4% (1,419) |
| average | 49.5% (1,496) | 47.0% (1,505) | 48.2% (3,001) |
| high | 24.2% (756) | 34.3% (1,095) | 29.4% (1,851) |
| **Negative attitude towards PLHIV** | | | |
| low level | 60.5% (1,867) | 66.2% (2,093) | 63.4% (3,960) |
| high level | 39.5% (1,201) | 33.8% (1,110) | 36.6% (2,311) |
| **EXPOSURE TO THE MEDIAS AND CONTEXTUAL FACTORS** | | | |
| **Exposure to the media** | | | |
| low | 71.0% (2,124) | 51.3% (1,560) | 61.0% (3,684) |
| high | 29.0% (932) | 48.7% (1,635) | 39.0% (2,567) |
| **Place of residence** | | | |
| rural | 59.9% (1,453) | 63.4% (1,608) | 61.7% (3,061) |
| urban | 40.1% (1,615) | 36.6% (1,595) | 38.3% (3,210) |
| **Department** | | | |
| San-Pedro | 35.0% (843) | 34.3% (838) | 34.6% (1,681) |
| Soubre | 23.8% (879) | 24.3% (908) | 24.1% (1,787) |
| Tabou | 7.9% (367) | 10.2% (469) | 9.1% (836) |
| Others | 33.2% (979) | 31.2% (988) | 32.2% (1,967) |

*(Continued)*

**Table 1.** (Continued)

| Characteristics | Women (n = 3068)¹ | Men (n = 3203)¹ | Overall (n = 6,274)¹ |
|---|---|---|---|
| OUTCOME VARIABLES | | | |
| **Already heard about HIVST knowledge** | | | |
| no | 89.9% (2,762) | 88.9% (2,863) | 89.4% (5,625) |
| yes | 10.1% (305) | 11.1% (338) | 10.6% (643) |
| **Interested in using HIVST for themselves** | | | |
| not interested | 27.3% (781) | 21.6% (679) | 24.4 (1,460) |
| interested/ very interested | 72.7% (2,255) | 78.4% (2,507) | 75.6% (4,762) |
| **Interested in using HIVST for sexual partners** | | | |
| not interested | 28.2% (806) | 22.3% (703) | 25.2% (1,509) |
| interested/ very interested | 71.8% (2,216) | 77.7% (2,472) | 74.8% (4,688) |
| **Already used HIVST** | | | |
| no | 97.4% (2,988) | 97.4% (3,130) | 97.4% (6,118) |
| yes | 2.6% (75) | 2.6% (69) | 2.6% (144) |

¹ *weighted % (unweighted n).*

## 3.3. Factors associated with attitudes towards HIVST

The participants' interest in using HIVST for their sexual partners if freely available in the future (column 3 in Table 2) appeared to be significantly associated with all independent variables, except the department and place of residence. In addition to the last two variables, the participants' interest in using HIVST for themselves in the future was not significantly associated with age group or being in a relationship.

Multivariate analysis (Fig 1 and columns 2 in S5 Table) showed that women with secondary or higher education (aOR:2.64 [95%CI: 1.75 to 3.99]), primary education (aOR:2.34 [95%CI: 1.77 to 3.10]), with one sexual partner (aOR:2.16 [95%CI: 1.50 to 3.09]), with two or more sexual partners (aOR:3.10 [95%CI: 1.65 to 5.83]) and low level of negative attitude towards PLHIV (aOR:1.96 [95%CI: 1.56 to 2.46]) had higher odds of being interested in using HIVST for themselves, while those with low HIV-related knowledge (aOR:0.38 [95%CI: 0.30 to 0.48]) had lower odds. On the other hand, compared to the reference groups, women who had secondary or higher education (aOR:3.00 [95%CI: 2.02 to 4.45]), primary education (aOR:2.60 [95%CI: 1.84 to 3.66]), one partner (aOR:2.66 [95%CI: 1.81 to 3.93]), 2 or more sexual partners (aOR:3.99 [95%CI: 1.94 to 8.21]) and low negative attitude towards PLHIV (aOR:1.92 [95%CI: 1.50 to 2.46]) were more likely to be interested in using HIVST for sexual partners. They were less likely to be interested when they had, a poor level of HIV knowledge (aOR:0.42 [95%CI: 0.33 to 0.52]) compared to those with moderate knowledge (see column 3 in S5 Table).

A similar pattern of association was found among men for both interest in using HIVST for themselves and for their sexual partners (see columns 2 and 3 in S4 Table and Fig 1).

## 3.4. Factors associated with HIVST practice

Bivariate analyses showed that the use of HIVST was positively associated with age (p = 0002), with the age group 25–34 having the highest proportion of HIVST use (4%). HIVST use was also positively associated with education (p = 0.031), wealth (p < 0.001), the number of sexual partners (p = 0.003), the level of HIV-related knowledge (p < 0.001), low negative attitude towards PLHIV (p < 0.001), exposure to the media (p < 0.001) and being in a relationship (p < 0.001) (see the fourth column in Table 2).

After controlling for other factors, in multivariable analysis, it appeared that men aged 25–34 (aOR:3.10 [95%CI: 1.65 to 5.80]) and from the wealthiest tercile (aOR:4.20 [95%CI: 1.58 to 11.2]) had a higher odd of already used HIVST than

**Table 2. Bivariate analysis of factors associated with KAP relating to HIVST among individuals aged 15-49 years in Côte d'Ivoire.**

| | Already heard about HIVST knowledge | | Interested in using HIVST for themselves | | Interested in using HIVST for sexual partners | | Already used HIVST | |
|---|---|---|---|---|---|---|---|---|
| | % (n)[1] | p-value[2] | % (n)[1] | p-value[2] | % (n)[1] | p-value[2] | % (n)[1] | p-value[2] |
| **Sex** | | 0.3 | | 0.002 | | <0.001 | | >0.9 |
| woman | 10.1% (305) | | 72.7% (2,255) | | 71.8% (2,216) | | 2.6% (75) | |
| man | 11.1% (338) | | 78.4% (2,507) | | 77.7% (2,472) | | 2.6% (69) | |
| **Age group** | | 0.035 | | 0.2 | | 0.029 | | 0.002 |
| 15-24 years old | 9.3% (233) | | 74.7% (1,846) | | 73.2% (1,782) | | 1.7% (42) | |
| 25-34 years old | 12.3% (214) | | 77.4% (1,444) | | 77.3% (1,452) | | 3.6% (59) | |
| 35-49 years old | 10.4% (196) | | 74.9% (1,472) | | 74.2% (1,454) | | 2.7% (43) | |
| **Level of education** | | <0.001 | | <0.001 | | <0.001 | | 0.031 |
| none | 6.1% (168) | | 66.1% (1,889) | | 65.1% (1,848) | | 1.7% (41) | |
| primary | 10.1% (132) | | 81.4% (1,096) | | 80.7% (1,084) | | 2.9% (32) | |
| secondary or higher | 17.7% (343) | | 85.6% (1,777) | | 85.0% (1,756) | | 3.8% (71) | |
| **Wealth index[3]** | | <0.001 | | 0.001 | | 0.003 | | <0.001 |
| poor | 6.0% (138) | | 70.3% (1,485) | | 68.9% (1,455) | | 1.1% (21) | |
| neither poor nor rich | 8.2% (136) | | 75.6% (1,401) | | 75.6% (1,397) | | 1.5% (19) | |
| rich | 17.8% (369) | | 81.4% (1,876) | | 80.4% (1,836) | | 5.3% (104) | |
| **Currently in a relationship** | | 0.009 | | 0.093 | | <0.001 | | <0.001 |
| no | 8.6% (139) | | 73.4% (1,172) | | 68.4% (1,086) | | 1.4% (23) | |
| yes | 11.2% (504) | | 76.3% (3,590) | | 76.7% (3,602) | | 3.0% (121) | |
| **Number of sexual partners over the last 12 months** | | <0.001 | | <0.001 | | <0.001 | | 0.003 |
| 0 partner | 7.6% (79) | | 66.6% (716) | | 60.9% (643) | | 1.5% (13) | |
| 1 partner | 9.7% (416) | | 76.5% (3,300) | | 76.6% (3,297) | | 2.5% (100) | |
| 2 partners or more | 17.9% (145) | | 82.0% (736) | | 81.9% (737) | | 4.3% (29) | |
| **The person who generally makes the decision about their own health care** | | 0.008 | | <0.001 | | <0.001 | | 0.004 |
| themselves | 12.3% (289) | | 79.2% (1,940) | | 78.9% (1,932) | | 2.9% (65) | |
| a third party | 8.7% (304) | | 72.9% (2,517) | | 71.7% (2,452) | | 2.0% (63) | |
| themselves and a third party | 15.9% (50) | | 76.0% (305) | | 75.0% (304) | | 5.8% (16) | |
| **HIV knowledge** | | <0.001 | | <0.001 | | <0.001 | | <0.001 |
| low | 4.9% (78) | | 57.2% (826) | | 56.6% (809) | | 0.6% (9) | |
| average | 10.5% (301) | | 78.5% (2,355) | | 77.1% (2,305) | | 2.8% (70) | |
| high | 15.0% (264) | | 84.6% (1,581) | | 84.6% (1,574) | | 3.9% (65) | |
| **Negative attitude towards PLHIV** | | <0.001 | | <0.001 | | <0.001 | | <0.001 |
| low level | 13.1% (506) | | 82.2% (3,291) | | 81.5% (3,253) | | 3.5% (120) | |
| high level | 6.3% (137) | | 64.0% (1,471) | | 63.1% (1,435) | | 1.2% (24) | |
| **Exposure to the media** | | <0.001 | | <0.001 | | <0.001 | | <0.001 |
| low | 7.3% (261) | | 71.8% (2,648) | | 71.2% (2,609) | | 1.7% (57) | |
| high | 15.8% (382) | | 81.6% (2,101) | | 80.5% (2,068) | | 4.1% (87) | |
| **Place of residence** | | 0.089 | | 0.15 | | 0.2 | | 0.3 |
| rural | 8.9% (250) | | 73.9% (2,251) | | 73.3% (2,229) | | 2.2% (53) | |
| urban | 13.3% (393) | | 78.3% (2,511) | | 77.3% (2,459) | | 3.4% (91) | |
| **Department** | | 0.10 | | 0.7 | | 0.6 | | 0.086 |

*(Continued)*

**Table 2.** (Continued)

| | Already heard about HIVST knowledge | | Interested in using HIVST for themselves | | Interested in using HIVST for sexual partners | | Already used HIVST | |
|---|---|---|---|---|---|---|---|---|
| | % (n)[1] | p-value[2] | % (n)[1] | p-value[2] | % (n)[1] | p-value[2] | % (n)[1] | p-value[2] |
| San-Pedro | 13.4% (210) | | 76.9% (1,259) | | 76.0% (1,234) | | 3.9% (53) | |
| Soubre | 10.1% (197) | | 73.0% (1,325) | | 71.2% (1,292) | | 2.3% (43) | |
| Tabou | 4.4% (44) | | 76.4% (644) | | 77.4% (652) | | 0.8% (11) | |
| Others | 9.7% (192) | | 76.0% (1,534) | | 75.5% (1,510) | | 2.0% (37) | |

[1] weighted % (unweighted n)

[2] Pearson's X^2: Rao & Scott adjustment

[3] The wealth index was calculated using multiple correspondence analysis (MCA) from household asset variables.

those aged 15–25 and from the poorest tercile. Among the women, only wealth was significantly associated with the use of HIVST, with women from the wealthiest tercile having odds of already used HIVST, 2.77 times greater than those from the poorest tercile [95% CI: 1.25 to 6.13]

## 4. Discussion and conclusion

This study aimed to assess levels and correlates of knowledge of, attitudes towards, and practices relating to HIV self-testing in the general population after its introduction by the ATLAS programme in Côte d'Ivoire. It drew on a representative survey carried out in the Bas-Sassandra region at the end of 2021.

The results showed that about 24 months after the introduction of HIVST in the region via the ATLAS programme, knowledge of HIVST remained relatively low overall (11%). The actual use of HIVST was even lower (3%). Although this proportion seems low, it is higher than results found by Terefe *et al.* among women in 21 Sub-Saharan African countries between 2015 and 2022. They found that prevalence of knowledge, and use of HIVST among women was about 2.17% (95% CI: 2.12–2.23) [23] – although the measures used by the authors were not really comparable to ours, as they did not enable a distinction between knowledge and use.

The levels of HIVST knowledge that we found in the study population seems surprisingly high, given the scale of the ATLAS programme. Indeed, the ATLAS initiatives were of rather modest proportions given the size of the population in the region – 74,785 HIVST kits were distributed over a three-year period on a territory of about 1,057,241 individuals aged 16–49 years. In addition, the programme activities were circumscribed, targeting key and vulnerable populations (and their partners), and especially those remote from the healthcare system but contributing substantially to the HIV spread because of the prevention gaps. About 90% of the ATLAS HIVST kits were distributed to key populations, accounting for less than 3% of the total population – about 1.4% of women for FSW and 1.3% of men for MSM in 2019 [22].

The restricted targeted population of the ATLAS programme was in line with national recommendations regarding priority populations, on the basis of prevalence rates. As a result, in order to avoid any stigmatization, since only a few specific groups had access to HIVST, the strategies for creating demand were designed to reach the targeted populations exclusively. Consequently, no communication on HIVST targeted the general population, which ultimately had an impact on the level of knowledge and use of HIVST observed in this study.

We found very high levels of positive attitudes towards HIVST, with three quarters of the respondents expressing willingness to use it for themselves and their sexual partners, if freely available. These results are in line with those found in ATLAS qualitative research among sex workers in West Africa, which found that they were willing to use and/or reuse HIVST and distribute it to regular clients and partners [19]. Similar proportions were found in Kenya and Rwanda, among young adults living in informal urban settlements and among male clinic attendees respectively [28,29]. In line with

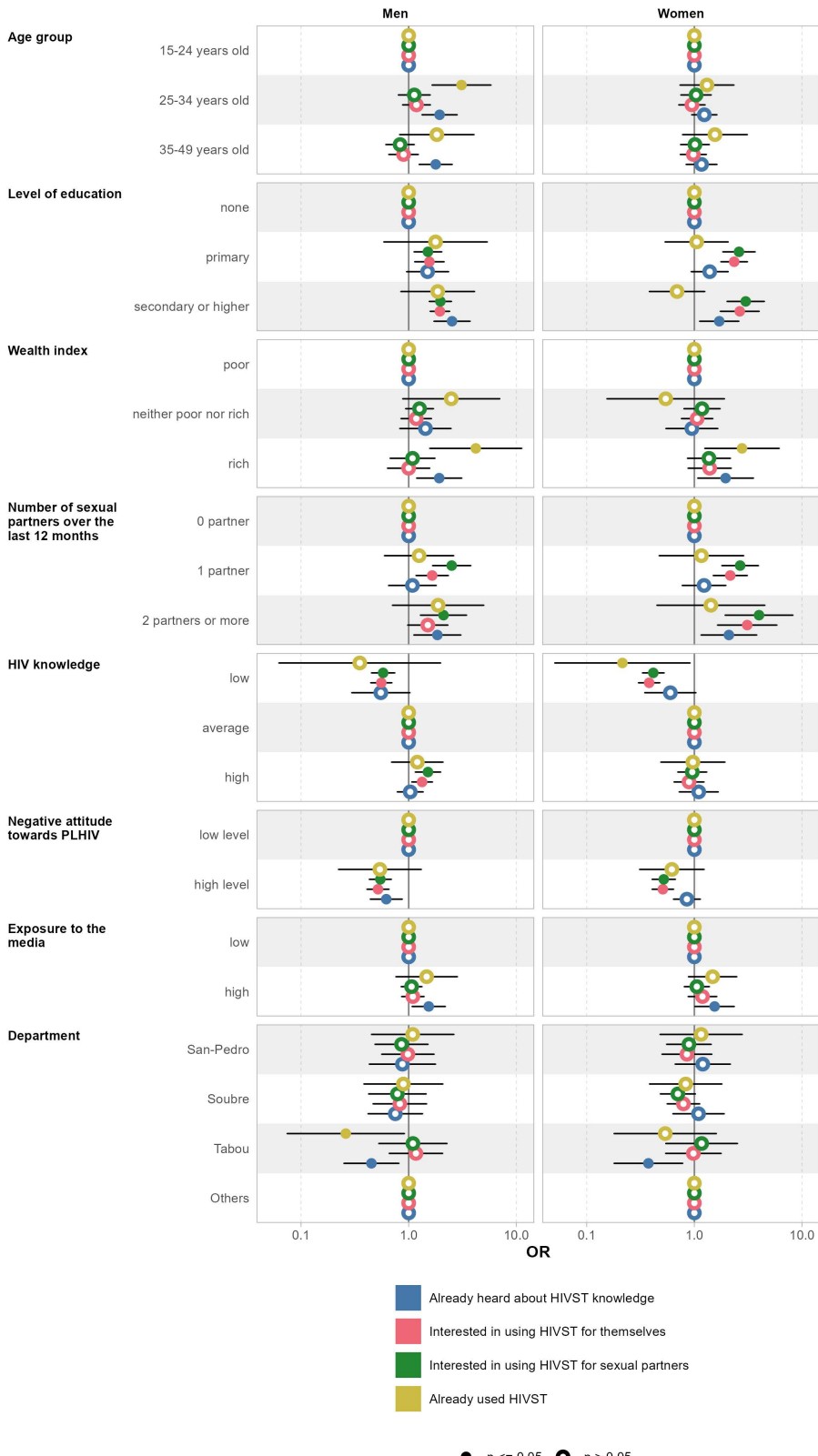

**Fig 1. Odds ratios from the logistic regression for KAP relating to HIVST in Côte d'Ivoire, stratified by sex (15-49 years old).**

our study, several surveys have shown a higher acceptance of HIVST among men than among women [13], probably because the former are usually more remote from the healthcare system than women who are likely to receive antenatal care. Confidentiality, empowerment, responsibility and saving of time and financial resources related to eventual transportation costs to HIV testing facilities are some of the reasons explaining the overall positive attitudes towards HIVST [17,30–33].

Multivariate analysis showed that exposure to the media appeared to be correlated with HIVST knowledge. This could be explained by the MTV Shuga multimedia campaign, which included an episode in its second season promoting HIVST. However, this result should be interpreted cautiously, because although we cannot rule out the possibility that other non-targeted age groups may have been exposed to it, the MTV-Shuga multimedia campaign was designed to target mostly the youngest people (aged 15–24 years), and our results showed that knowledge, attitudes and practices in this age group were no better than in the older age group. In fact, the opposite is true. Exposure to the media (TV, Internet and the radio) could be a proxy for better general culture.

The place of residence did not appear to be strongly correlated with KAP relating to HIVST despite differences in ATLAS activities intensity. This could be partly due to the fact that the ATLAS activities were fairly modest and targeted, as previously explained. On the other hand, because of population mobility across departments, information and HIVST kits could also have circulated across departmental boundaries, making it difficult to observe any difference between departments initially targeted by the ATLAS programme activities and those receiving almost no distribution of kits.

Regarding the socio-demographic and socio-economic factors, education was positively associated with knowledge and positive attitudes towards HIVST for both men and women. Wealth was positively associated with knowledge and practices concerning HIVST in the two groups. This corroborates previous evidence in the literature on Sub-Saharan African countries and China [23,34–36]. It is usually explained by the fact that education and wealth are associated with knowledge of HIV status, use of healthcare services, and health-seeking behaviours. Age appeared to be positively correlated with knowledge and use of HIVST among the men only. Previous research has found that age is positively associated with HIVST knowledge and use for both genders [23,28]. This could be explained by life experiences or personal levels of risk-taking, but also by higher costs of going to a healthcare facility to undergo a standard HIV test for older adults.

Behavioural factors were also associated with KAP relating to HIVST, with a positive association found between the number of sexual partners over the last 12 months and the participants' knowledge of HIVST and willingness to use it for themselves or for their sexual partners among both men and women. The effect was even stronger for those with two or more sexual partners, corroborating studies that have shown that individuals with risky sexual behaviours are more likely to know and accept HIVST [29]. Contrasted results were found in other studies, although they focused on different, younger populations than the one in this study [28,37].

We also found that, if HIV-related knowledge and attitude did not appear to be associated with knowledge and practice of HIVST, especially among the women, it was an important correlate of attitudes towards HIVST for both men and women. This could be explained by the fact that a good knowledge of HIV generally implies awareness of the benefits of knowing one's own HIV status, while low stigma or negative attitude toward PLHIV reduces their fear of receiving a positive HIV test result, making them more willing to use HIVST if available to them. In fact, the literature suggests that HIV-related stigma is an important barrier to testing [38]. Previous studies have also found a significant association between having negative attitudes towards HIVST and high HIV-related stigma, though this association did not show in multivariable analysis [39].

This work presents some limitations. Firstly, because of budget limitations, the survey was restricted to, and is representative of, only one region in Côte d'Ivoire, making it difficult to extrapolate the results to other regions or countries. However, there is no serious reason to think that our results would be completely different from those that might be observed in other Ivorian regions. Secondly, the sample size did not allow for a specific analysis of key populations or the ATLAS targeted groups. However, some qualitative studies and a telephone survey targeting these groups have already been conducted as part of the ATLAS project, and they have shown that HIVST is very well accepted, particularly by key

populations, and does indeed reach people living with HIV [16,19,40]. The cross-sectional nature of the survey makes it difficult to establish a cause-and-effect relationship, so future studies could adopt a study design that addresses this issue.

Nevertheless, the current study contributes to the literature in different ways. Firstly, it enabled an assessment, at population level, of knowledge of, attitudes towards, and practices relating to HIVST, following a programme introducing HIVST via targeted populations, allowing some spill-over effects to be captured. Secondly, unlike some previous studies, it assessed the levels of HIVST knowledge, attitudes and use among men and women separately, as well as their correlates, enabling more detailed policy recommendations.

The study suggests, for example, that actions aimed at improving general HIV-related knowledge and reducing negative attitude toward PLHIVcould help improve attitudes toward HIVST. Additionally, improving access to the media and providing more information about HIVST through these channels could enhance awareness. Less educated and poorer groups have the lowest levels of KAP relating to HIVST and should therefore be the focus of policymakers. The high level of positive attitudes towards HIVST calls for a scaling-up of access to HIVST in the region through dedicated programmes or through its availability in private pharmacies at low prices. Working along these lines could help policymakers improve HIV testing and progress towards UNAIDS targets of achieving 95% of PLHIV knowing their status by 2025.

## Supporting information

**S1 Checklist. Inclusivity in global research questionnaire.**
(DOCX)

**S1 Table. Questions about HIV self-testing Knowledge, attitude and practice.**
(DOCX)

**S2 Table. Questions about HIV-related knowledge used to construct the knowledge score.**
(DOCX)

**S3 Table. HIV-related attitude questions used to construct the negative attitude score.**
(DOCX)

**S4 Table. Odds ratios from the multivariable logistic regression for KAP relating to HIVST among men aged 15–49 years in Côte d'Ivoire.**
(DOCX)

**S5 Table. Odds ratios from the multivariable logistic regression for KAP relating to HIVST among women aged 15–49 years in Côte d'Ivoire.**
(DOCX)

## Acknowledgments

Ethics approvals were obtained from the Côte d'Ivoire Ministry of Health (N/ref: 051–21/MSHP/CNESVS-km), the London School of Hygiene and Tropical Medicine (LSHTM) (LSHTM Ethics Ref: 26258), and the World Health Organization (WHO) (ERC.0003596) ethics committees in June, July and September 2021 respectively. The authors would like to thank all participants in this study, all the people involved in the data collection process, and the field workers in Côte d'Ivoire. We would also like to thank the members of the ATLAS Team lead by Joseph Larmarange (joseph.larmarange@ird.fr).

Composition of the ATLAS Team:
***ATLAS Research Team***
Amani Elvis Georges (Programme PACCI, ANRS Research Site, Treichville University Hospital,Abidjan, Côte d'Ivoire); Badiane Kéba (Solthis, Sénégal); Bayac Céline (Solthis, France); Bekelynck Anne (Programme PACCI, ANRS Research

Site, Treichville University Hospital, Abidjan, Côte d'Ivoire); Boily Marie-Claude (Department of Infectious Disease Epidemiology, Medical Research Council Centre for Global Infectious Disease Analysis, Imperial College London, London, United Kingdom); Boye Sokhna (Centre Population et Développement, Institut de Recherche pour le Développement, Université Paris Descartes, Inserm, Paris, France); Breton Guillaume (Solthis, Paris, France); d'Elbée Marc (Department of Global Health and Development, Faculty of Public Health and Policy, London School of Hygiene and Tropical Medicine, London, United Kingdom); Desclaux Alice (Institut de Recherche pour le Développement, Transvihmi (UMI 233 IRD, 1175 INSERM, Montpellier University), Montpellier, France/CRCF, Dakar, Sénégal); Desgrées du LoÛ Annabel (Centre Population et Développement, Institut de Recherche pour le Développement, Université Paris

Descartes, Inserm, Paris, France); Diop Papa Moussa (Solthis, Sénégal); Ehui Eboi (Directeur

Coordonnateur, PNLS; Medley Graham, Department of Global Health and Development, Faculty of Public Health and Policy, London School of Hygiene and Tropical Medicine, London, United Kingdom); Jean Kévin (Laboratoire MESuRS, Conservatoire National des Arts et Métiers, Paris, France); Keita Abdelaye (Institut National de Recherche en Santé Publique, Bamako, Mali); Kouassi Arsène Kra (Centre Population et Développement, Institut de Recherche pour le Développement, Université Paris Descartes, Inserm, Paris, France); Ky-Zerbo Odette (TransVIHMI, IRD, Université de Montpellier, INSERM); Larmarange Joseph (Centre Population et Développement, Institut de Recherche pour le Développement, Université Paris Descartes, Inserm, Paris, France); Maheu-Giroux Mathieu (Department of Epidemiology, Biostatistics, and Occupational Health, School of Population and Global Health, McGill University, Montréal, QC, Canada); Moh Raoul (Programme PACCI, ANRS Research Site, Treichville University Hospital, Abidjan, Côte d'Ivoire; Department of Infectious and Tropical Diseases, Treichville University Teaching Hospital, Abidjan, Côte d'Ivoire; Medical School, University Felix Houphouet Boigny, Abidjan, Côte d'Ivoire); Mosso Rosine (ENSEA Ecole Nationale de Statistiques et d'Economie Appliquée, Abidjan, Côte d'Ivoire); Ndour Cheikh Tidiane (Division de Lutte contre le Sida et les IST, Ministère de la Santé et de l'Action Sociale Institut d'Hygiène Sociale, Dakar, Sénégal); Paltiel David (Yale School of Public Health, New Haven, CT, United States); Pourette Dolorès (Centre Population et Développement, Institut de Recherche pour le Développement, Université Paris Descartes, Inserm, Paris, France); Rouveau Nicolas (Centre Population et Développement, Institut de Recherche pour le Développement, Université Paris Descartes, Inserm, Paris, France); Silhol Romain (Medical Research Council Centre for Global Infectious Disease Analysis, Department of Infectious Disease Epidemiology, Imperial College London, London, United Kingdom); Simo Fotso Arlette (Centre Population et Développement, Institut de Recherche pour le Développement, Université Paris Descartes, Inserm, Paris, France); Terris-Prestholt Fern (Department of Global Health and Development, Faculty of Public Health and Policy, London School of Hygiene and Tropical Medicine, London, United Kingdom); Traore Métogara Mohamed (Solthis, Côte d'Ivoire).

***Solthis Coordination Team***

Diallo Sanata (Solthis, Dakar, Sénégal); Doumenc-Aïdara Clémence (Solthis, Dakar, Sénégal);

Geoffroy Olivier (Solthis, Abidjan, Côte d'Ivoire); Kanku Kabemba Odé (Solthis, Bamako, Mali);

Vautier Anthony (Solthis, Dakar, Sénégal)

***Implementation in Côte d'Ivoire***

Abokon Armand (Fondation Ariel Glaser, Côte d'Ivoire); Anoma Camille (Espace Confiance, Côte d'Ivoire); Diokouri Annie (Fondation Ariel Glaser, Côte d'Ivoire); Kouame Blaise (Service Dépistage, PNLS); Kouakou Venance (Heartland Alliance, Côte d'Ivoire); Koffi Odette (Aprosam, Côte d'Ivoire); Kpolo Alain (Michel-Ruban Rouge, Côte d'Ivoire); Tety Josiane (Blety, Côte d'Ivoire); Traore Yacouba (ORASUR, Côte d'Ivoire).

***Implementation in Mali***

Bagendabanga Jules (FHI 360, Mali); Berthé Djelika (PSI, Mali); Diakite Daouda (Secrétariat

Exécutif du Haut Conseil National de Lutte contre le Sida, Mali); Diakité Mahamadou (Danayaso, Mali); Diallo Youssouf (CSLS/MSHP); Daouda Minta (Comité scientifique VIH); Hessou Septime (Plan Mali); Kanambaye Saidou (PSI, Mali); Kanoute Abdul Karim (Plan Mali); Keita Dembele Bintou (Arcad-Sida, Mali); Koné Dramane (Secrétariat Exécutif du Haut

Conseil National de Lutte contre le Sida, Mali); Koné Mariam (AKS, Mali); Maiga Almoustapha (Comité scientifique VIH; Nouhoum Telly, CSLS/MSHP); Saran Keita Aminata (Soutoura, Mali); Sidibé Fadiala (Soutoura, Mali); Tall Madani (FHI 360, Mali); Yattassaye Camara Adam (Arcad-Sida, Mali); Sanogo Abdoulaye (Amprode Sahel, Mali).

*Implementation in Senegal*

Bâ Idrissa (CEPIAD, Sénégal); Diallo Papa Amadou Niang (CNLS, Sénégal); Fall Fatou (DLSI, Ministère de la Santé et de l'action sociale, Sénégal); Guèye NDèye Fatou NGom (CTA, Sénégal); Ndiaye Sidy Mokhtar (Enda Santé, Sénégal); Niang Alassane Moussa (DLSI, Ministère de la Santé et de l'action sociale, Sénégal); Samba Oumar (CEPIAD, Sénégal); Thiam Safiatou (CNLS, Sénégal); Turpin Nguissali M.E. (Enda Santé, Sénégal).

*Partners*

Bouaré Seydou (Assistant de recherche, Mali); Camara Cheick Sidi (Assistant de recherche, Mali); Kouadio Brou Alexis (Assistant de recherche, Côte d'Ivoire); Sarrassat Sophie (Centre for Maternal, Adolescent, Reproductive and Child Health, London School of Hygiene and Tropical Medicine, London, United kingdom); Sow Souleyman (Assistant de recherche, Sénégal).

## Author contributions

**Conceptualization:** Arlette Simo Fotso, Joseph Larmarange.

**Data curation:** Christian Koukobo.

**Formal analysis:** Arlette Simo Fotso, Christian Koukobo, Joseph Larmarange.

**Funding acquisition:** Anthony Vautier, Joseph Larmarange.

**Investigation:** Arlette Simo Fotso.

**Methodology:** Arlette Simo Fotso, Joseph Larmarange.

**Project administration:** Anthony Vautier.

**Supervision:** Arlette Simo Fotso.

**Validation:** Arlette Simo Fotso, Joseph Larmarange.

**Visualization:** Joseph Larmarange.

**Writing – original draft:** Arlette Simo Fotso.

**Writing – review & editing:** Christian Koukobo, Romain Silhol, Arsène Kouassi Kra, Marie-Claude Boily, Anthony Vautier, Joseph Larmarange.

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
