## [Decision Letter · Decision Letter 0]

21 Feb 2025

Dear Dr. Larmarange,

We look forward to receiving your revised manuscript.

Kind regards,

Leila Molaeipour

Academic Editor

PLOS ONE

Journal Requirements:

2. Please include a complete copy of PLOS’ questionnaire on inclusivity in global research in your revised manuscript. Our policy for research in this area aims to improve transparency in the reporting of research performed outside of researchers’ own country or community. The policy applies to researchers who have travelled to a different country to conduct research, research with Indigenous populations or their lands, and research on cultural artefacts. The questionnaire can also be requested at the journal’s discretion for any other submissions, even if these conditions are not met.

Please find more information on the policy and a link to download a blank copy of the questionnaire here: https://journals.plos.org/plosone/s/best-practices-in-research-reporting.

Please upload a completed version of your questionnaire as Supporting Information when you resubmit your manuscript.

“This analysis was supported by Unitaid (grant number 2018–23 ATLAS) through a collaborative agreement with Solthis.”

4. Please note that funding information should not appear in the Acknowledgments section or other areas of your manuscript. We will only publish funding information present in the Funding Statement section of the online submission form. Please remove any funding-related text from the manuscript.

5. We note that you have indicated that there are restrictions to data sharing for this study. For studies involving human research participant data or other sensitive data, we encourage authors to share de-identified or anonymized data. However, when data cannot be publicly shared for ethical reasons, we allow authors to make their data sets available upon request. For information on unacceptable data access restrictions, please see http://journals.plos.org/plosone/s/data-availability#loc-unacceptable-data-access-restrictions.

6. In the online submission form, you indicated that:

“Data cannot be shared publicly at this time, as it was collected through a joint survey conducted by the ATLAS program and the MTV Shuga initiative. As members of the ATLAS program team, we can only share the data once MTV Shuga has published results on the primary outcomes of their project. In the meantime, the data will be available upon request, and we have included it as a supplementary document during the submission process for reviewers’ reference. Ultimately, the data will be made publicly available on Zenodo.”

7. One of the noted authors is a group or consortium: ATLAS team

In addition to naming the author group, please list the individual authors and affiliations within this group in the acknowledgments section of your manuscript. Please also indicate clearly a lead author for this group along with a contact email address.

8. We are unable to open your Supporting Information file data.zip. Please kindly revise as necessary and re-upload.

**Additional Editor Comments:**

Investigating knowledge about HIV self-testing is crucial for strengthening public health strategies to increase testing rates and reduce HIV-related stigma. While our study provides significant evidence in this area, there are several key issues that still need to be addressed.

1- Include keywords derived from MeSH terms.

2- The introduction does not transition smoothly from general concepts to specific details. Please revise the introduction to ensure clarity and maintain coherence.

3- Please include the sample size formula as well as the criteria for inclusion and exclusion in the study.

4- Please provide the content of the knowledge, attitude, and practice survey form, including the number of questions and the scoring method used.

5- In the methods section on page 6, it is unclear whether HIV-related knowledge and stigma or attitude have been investigated. Please clarify, as attitude and stigma are distinct concepts and should be addressed separately.

6- Please include the Ethics Committee Code in the manuscript.

Reviewers' comments:

Reviewer's Responses to Questions

**Comments to the Author**

1. Is the manuscript technically sound, and do the data support the conclusions?

Reviewer #1: Yes

Reviewer #2: No

2. Has the statistical analysis been performed appropriately and rigorously?

Reviewer #1: Yes

Reviewer #2: No

3. Have the authors made all data underlying the findings in their manuscript fully available?

Reviewer #1: No

Reviewer #2: Yes

4. Is the manuscript presented in an intelligible fashion and written in standard English?

Reviewer #1: Yes

Reviewer #2: Yes

Reviewer #1: This manusccrips describes apect knowlledge attitude and practices towrads HIV self testing the Bas-Sassandra region of Côte d’Ivoire. The reserach questions were carefully enunciateds, the design and sampling methods were sound and the analystical techniques employed were appropriate considering the design and research questions.

Overall the manuscript was well written and it was joy to read. The authors report on some signoificant findings which although not novell does provide the foundation to faciliatate evaluation of programme effficay should scaling be pursued

Reviewer #2: 1. Keyword based on mesh is not

2. The introduction does not proceed clearly from generalities to details.

3. Sample size formula?

4. Study inclusion and exclusion criteria?

5. Content of the knowledge, attitude, and practice survey form (number of questions) and its scoring method.

6. Is it attitude or stigma that has been investigated?

7. Attitude and stigma are two separate concepts.

8. Ethics Committee Code?

**Do you want your identity to be public for this peer review?** For information about this choice, including consent withdrawal, please see our Privacy Policy

Reviewer #1: **Yes:** MARVIN REID

Reviewer #2: No

---

## [Author Response · Author response to Decision Letter 1]

7 Apr 2025

We thank the Editor and reviewers for their helpful comments, which have improved the manuscript. Below, you will find our responses in purple to each point raised by the academic editor and reviewers.

Editor Comments

Journal Requirements:

The manuscript style has been updated accordingly

2. Please include a complete copy of PLOS’ questionnaire on inclusivity in global research in your revised manuscript. Our policy for research in this area aims to improve transparency in the reporting of research performed outside of researchers’ own country or community. The policy applies to researchers who have travelled to a different country to conduct research, research with Indigenous populations or their lands, and research on cultural artefacts. The questionnaire can also be requested at the journal’s discretion for any other submissions, even if these conditions are not met.

Please find more information on the policy and a link to download a blank copy of the questionnaire here: https://journals.plos.org/plosone/s/best-practices-in-research-reporting.

Please upload a completed version of your questionnaire as Supporting Information when you resubmit your manuscript.

The questionnaire has been uploaded.

“This analysis was supported by Unitaid (grant number 2018–23 ATLAS) through a collaborative agreement with Solthis.”

The cover letter has been amended with the following statement “This analysis was supported by Unitaid (grant number 2018–23 ATLAS) through a collaborative agreement with Solthis. The funders had no role in study design, data collection and analysis, decision to publish, or preparation of the manuscript.”

4. Please note that funding information should not appear in the Acknowledgments section or other areas of your manuscript. We will only publish funding information present in the Funding Statement section of the online submission form. Please remove any funding-related text from the manuscript.

The funding-related information has been removed from the acknowledgements as suggested

5. We note that you have indicated that there are restrictions to data sharing for this study. For studies involving human research participant data or other sensitive data, we encourage authors to share de-identified or anonymized data. However, when data cannot be publicly shared for ethical reasons, we allow authors to make their data sets available upon request. For information on unacceptable data access restrictions, please see http://journals.plos.org/plosone/s/data-availability#loc-unacceptable-data-access-restrictions.

The minimal anonymized dataset and R script used to replicate the presented analyses have been made publicly available on Zenodo with the DOI: 10.5281/zenodo.15064708 .

The full survey data are considered only pseudonymized according to the European General Data Protection Regulation (GDPR) and therefore cannot be made publicly available. They can only be shared under specific restrictions by contacting the corresponding authors.

Data Availability statement has been updated accordingly.

6. In the online submission form, you indicated that:

“Data cannot be shared publicly at this time, as it was collected through a joint survey conducted by the ATLAS program and the MTV Shuga initiative. As members of the ATLAS program team, we can only share the data once MTV Shuga has published results on the primary outcomes of their project. In the meantime, the data will be available upon request, and we have included it as a supplementary document during the submission process for reviewers’ reference. Ultimately, the data will be made publicly available on Zenodo.”

The minimal anonymized dataset and R script used to replicate the presented analyses have been made publicly available on Zenodo with the DOI: 10.5281/zenodo.15064708 . See response to comment 6

7. One of the noted authors is a group or consortium: ATLAS team

In addition to naming the author group, please list the individual authors and affiliations within this group in the acknowledgments section of your manuscript. Please also indicate clearly a lead author for this group along with a contact email address.

Individuals from the ATLAS group have been named in the acknowledgments. Note that they do not meet the criteria for authorship of this paper; that is why we used “on behalf of the…” instead of “and the…” in the author list.

8. We are unable to open your Supporting Information file data.zip. Please kindly revise as necessary and re-upload.

The minimal anonymized dataset and R script used to replicate the presented analyses have been made publicly available on Zenodo with the DOI: 10.5281/zenodo.15064708 . See response to comments 6 and 7

Our reference list is complete and correct.

Additional Editor Comments:

Investigating knowledge about HIV self-testing is crucial for strengthening public health strategies to increase testing rates and reduce HIV-related stigma. While our study provides significant evidence in this area, there are several key issues that still need to be addressed.

1- Include keywords derived from MeSH terms.

The following keywords derived from MeSH were included: Health Knowledge, Attitudes, Practice; Self-Testing; HIV Infections; Humans, Cote d’Ivoire; Cross-Sectional Studies

2- The introduction does not transition smoothly from general concepts to specific details. Please revise the introduction to ensure clarity and maintain coherence.

The introduction was revised accordingly

3- Please include the sample size formula as well as the criteria for inclusion and exclusion in the study.

The following paragraph was added to the method section:

“The sample size was calculated to ensure that the study's power was estimated at 92% and 99% to detect a minimum difference of 10% and 15%, respectively, in the ATLAS primary outcome compared to the 2018 PHIA (Population-Based HIV Impact Assessment) survey. The inclusion criteria were individuals aged 15 to 49 years, de facto household members (i.e., those present in the household at the time of the survey), individuals aged 18 or older with written informed consent, and those aged 15–17 with written informed assent and parent/guardian's written informed consent. The exclusion criteria included individuals younger than 15 or older than 49 years, those not present in the household at the time of the survey, refusal by the participant and/or parent/guardian, refusal by individuals aged 18 or older, refusal by a parent/guardian or lack of assent from individuals aged 15–17, and cognitive issues preventing the individual from providing informed consent.”

4- Please provide the content of the knowledge, attitude, and practice survey form, including the number of questions and the scoring method used.

The questions about HIV self-testing (HIVST) knowledge, attitude, and practice are now provided in Supporting Information S1 Table. No special scoring method was used, as these were binary variables. To clarify how these variables were coded, the manuscript has been amended as follows:

“We defined four dichotomous dependent variables related to knowledge, attitudes, and practices relating to HIV self-testing derived from the questions presented in the supporting information S1 Table. The first dependent variable measured individuals’ knowledge of HIVST, assessing whether the respondent had already heard of HIV self-testing, coded as “yes” or “no”. The second and third dependent variables concerned the respondents’ interest in using HIVST for themselves or for their partners if it was freely available. Each of these were binary variables indicating whether the individuals responded that they interested /very interested or not. These variables reflected the respondents’ attitudes towards HIVST. The fourth variable reflecting practices asked individuals whether they had ever used HIVST. This was also a binary variable, coded as “yes” or “no”.”

5- In the methods section on page 6, it is unclear whether HIV-related knowledge and stigma or attitude have been investigated. Please clarify, as attitude and stigma are distinct concepts and should be addressed separately.

In the literature, HIV-related stigma is often seen as an umbrella concept and "refers to beliefs and/or attitudes about HIV" (see the articles doi: 10.1002/jia2.25915 or doi: 10.1007/s10461-009-9593-3). However, to address the reviewer's suggestion and given that our specific questions are mainly related to attitudes towards PLHIV, we have changed the wording to "negative attitudes toward PLHIV.

6- Please include the Ethics Committee Code in the manuscript.

Ethics committee codes have been included in the manuscript page 6 as follow “Ethics approvals were obtained from the Côte d’Ivoire Ministry of Health (N/ref: 051-21/MSHP/CNESVS-km), the London School of Hygiene and Tropical Medicine (LSHTM) (LSHTM Ethics Ref: 26258), and the World Health Organization (WHO) (ERC.0003596) ethics committees in June, July and September 2021 respectively.” It was also added to acknowledgement as per editor’s suggestion

Reviewer #1:

This manusccrips describes apect knowlledge attitude and practices towrads HIV self testing the Bas-Sassandra region of Côte d’Ivoire. The reserach questions were carefully enunciateds, the design and sampling methods were sound and the analystical techniques employed were appropriate considering the design and research questions.

Overall the manuscript was well written and it was joy to read. The authors report on some signoificant findings which although not novell does provide the foundation to faciliatate evaluation of programme effficay should scaling be pursued

We thank the reviewer for their very positive comments regarding the paper.

Reviewer #2:

1. Keyword based on mesh is not

Comment addressed. Please see our response to editor’s additional comment number 1.

2. The introduction does not proceed clearly from generalities to details.

Comment addressed. Please see our response to editor’s comment number 2.

3. Sample size formula?

Comment addressed. Please see our response to editor’s additional comment number 3.

4. Study inclusion and exclusion criteria?

Comment addressed. Please see our response to editor’s additional comment number 3.

5. Content of the knowledge, attitude, and practice survey form (number of questions) and its scoring method.

Comment addressed. Please see our response to editor’s additional comment number 4.

6. Is it attitude or stigma that has been investigated?

Comment addressed. Please see our response to editor’s additional comment number 5.

7. Attitude and stigma are two separate concepts.

Comment addressed. Please see our response to editor’s additional comment number 6.

8. Ethics Committee Code?

Comment addressed. Please see our response to editor’s additional comment number 7.

---

## [Decision Letter · Decision Letter 1]

13 Oct 2025

Dear Dr. Larmarange,

Thank you for submitting your manuscript to PLOS ONE. After careful consideration, we feel that it has merit but does not fully meet PLOS ONE’s publication criteria as it currently stands. Therefore, we invite you to submit a revised version of the manuscript that addresses the points raised during the review process.

We look forward to receiving your revised manuscript.

Kind regards,

Lara Vojnov

Academic Editor

PLOS ONE

Journal Requirements:

Reviewers' comments:

Reviewer's Responses to Questions

**Comments to the Author**

Reviewer #1: All comments have been addressed

Reviewer #3: All comments have been addressed

2. Is the manuscript technically sound, and do the data support the conclusions?

Reviewer #1: Yes

Reviewer #3: Yes

3. Has the statistical analysis been performed appropriately and rigorously?

Reviewer #1: Yes

Reviewer #3: I Don't Know

4. Have the authors made all data underlying the findings in their manuscript fully available?

Reviewer #1: Yes

Reviewer #3: Yes

5. Is the manuscript presented in an intelligible fashion and written in standard English?

Reviewer #1: Yes

Reviewer #3: Yes

Reviewer #1: (No Response)

Reviewer #3: The authors have addressed prior reviewer comments.

However one area that needs addressing, is the abstract - there is no indication of the number of people interviewed, where they came from, or how they were selected - a sentence or two on this would make all the difference.

**Do you want your identity to be public for this peer review?** For information about this choice, including consent withdrawal, please see our Privacy Policy

Reviewer #1: **Yes:** Marvin Reid

Reviewer #3: No

---

## [Author Response · Author response to Decision Letter 2]

27 Oct 2025

We thank the reviewers and the editor for their very positive comments.

Only point 6 required a response from us: However one area that needs addressing, is the abstract - there is no indication of the number of people interviewed, where they came from, or how they were selected - a sentence or two on this would make all the difference.

We have addressed this comment by adding the following sentences to the abstract: “A total of 6,271 people (3,203 men and 3,068 women) were interviewed. They were selected using a three-stage stratified sampling approach in the Bas-Sassandra region.”

---

## [Decision Letter · Decision Letter 2]

5 Jan 2026

Knowledge, attitudes and practices relating to HIV self-testing following its introduction in the Bas-Sassandra region of Côte d’Ivoire: the case of the ATLAS project

PONE-D-24-50776R2

Dear Dr. Larmarange,

We’re pleased to inform you that your manuscript has been judged scientifically suitable for publication and will be formally accepted for publication once it meets all outstanding technical requirements.

Kind regards,

Matthew J. Mimiaga, ScD, MPH, MA

Academic Editor, PLOS One

Additional Editor Comments (optional):

The study assesses knowledge, attitudes, and practices (KAP) regarding HIV self-testing (HIVST) in the Bas-Sassandra region of Côte d’Ivoire after the ATLAS program distributed a large number of HIVST kits. Using a 2021 cross-sectional survey of 6,271 individuals aged 15–49 with a three-stage stratified sampling design, the authors employed bivariate analyses and multivariable logistic regressions to examine correlates of KAP. They found that actual awareness of HIVST was low (11%) and recent use was even lower (3%), yet an overwhelming majority expressed strong interest in using HIVST if it were freely available (about 76% for personal use and 75% for their sexual partners). Education and wealth were consistently positively associated with better knowledge and more favorable attitudes for both men and women, while age showed a positive association with knowledge and use among men only. A higher number of sexual partners in the past year correlated with greater knowledge and willingness to use HIVST for oneself or one’s partners in both sexes. Media exposure was linked to greater knowledge, and higher HIV-related knowledge coupled with lower negative attitudes predicted more positive attitudes toward HIVST. The findings imply a large unmet demand for HIVST that could be leveraged if the service were freely available, with uptake shaped by socioeconomic status, sexual behavior, and media exposure. Overall, the paper is strong: it uses a large, regionally representative sample and robust analyses to illuminate KAP gaps and drivers, contributing valuable, actionable insights to the literature on HIVST uptake in West Africa. It is also responsive to prior reviews. As such, I think the paper should be accepted for publication.

Reviewers' comments:

Reviewer's Responses to Questions

**Comments to the Author**

Reviewer #1: All comments have been addressed

2. Is the manuscript technically sound, and do the data support the conclusions?

Reviewer #1: Yes

3. Has the statistical analysis been performed appropriately and rigorously?

Reviewer #1: Yes

4. Have the authors made all data underlying the findings in their manuscript fully available?

Reviewer #1: Yes

5. Is the manuscript presented in an intelligible fashion and written in standard English?

Reviewer #1: Yes

Reviewer #1: (No Response)

**Do you want your identity to be public for this peer review?** For information about this choice, including consent withdrawal, please see our Privacy Policy

Reviewer #1: **Yes:** Marvin Reid

---

## [Editor Report · Acceptance letter]

PONE-D-24-50776R2

PLOS One

Dear Dr. Larmarange,

I'm pleased to inform you that your manuscript has been deemed suitable for publication in PLOS One. Congratulations! Your manuscript is now being handed over to our production team.

Kind regards,

on behalf of

Dr. Matthew J. Mimiaga

Academic Editor

PLOS One